# Comparison of monocyte distribution width and Procalcitonin as diagnostic markers for sepsis: Meta-analysis of diagnostic test accuracy studies

**Karam R. Motawea[1], Samah S. Rozan[1], Nesreen Elsayed Talat[1], Rowan H. Elhalag[1], Sarraa Mohammed Reyad[1], pensée chebl[1], Sarya Swed [2]\*, Bisher Sawaf[3], Hadeel Hadeel alfar[3], Amr Farwati [3], Bana Sabbagh[4], Esperance M. Madera[5], Amro El Metaafy[6], Joshuan J. Barboza [7]\*, Ranjit Sah[8,9,10], Hani Aiash[11,12,13]**

1 Faculty of Medicine, Alexandria University, Alexandria, Egypt, 2 Faculty of Medicine, Aleppo University, Aleppo, Syria, 3 Internal Medicine Department, Hamad Medical Corporation, Doha, Qatar, 4 Department of Internal Medicine, Al-Mouwasat University Hospital, Al Mazzeh, Damascus, Syria, 5 University of Medicine and Health Sciences, 6 Faculty of Medicine, Suez Canal University, Ismailia, Egypt, 7 Escuela de Medicina, Universidad Cesar Vallejo, Trujillo, Peru, 8 Institute of Medicine, Tribhuvan University Teaching Hospital, Kathmandu, Nepal, 9 Department of Microbiology, Dr. D. Y. Patil Medical College, Hospital and Research Centre, Dr. D. Y. Patil Vidyapeeth, Pune, Maharashtra, India, 10 Department of Public Health Dentistry, Dr. D.Y. Patil Dental College and Hospital, Dr. D.Y. Patil Vidyapeeth, Pune, Maharashtra, India, 11 Department of Medicine, Upstate Medical University, Syracuse, NY, United States of America, 12 Department of Surgery, Upstate Medical University, Syracuse, NY, United States of America, 13 Department of Family Medicine, College of Medicine, Suez Canal University, Ismailia, Egypt

\* saryaswed1@gmail.com (SS); jbarbozame@ucvvirtual.edu.pe (JJB)

**Data Availability Statement:** All relevant data are within the paper.

## Abstract

### Aim

We aimed to perform a meta-analysis to find out whether PCT and MDW could be used as accurate diagnostic markers for sepsis.

### Methods

We searched PUBMED, WOS, and SCOPUS databases. Inclusion criteria were any observational or clinical trials that compared monocyte Distribution Width [MDW] with Procalcitonin [PCT] as diagnostic markers in a patient with sepsis. Case reports, editorials, conference abstracts, and animal studies were excluded. RevMan software [5.4] was used to perform the meta-analysis.

### Results

After the complete screening, 5 observational studies were included in the meta-analysis. The total number of patients included in the meta-analysis in the sepsis group is 565 and 781 in the control group. The pooled analysis between the sepsis group and controls showed a statistically significant association between sepsis and increased levels of MDW and PCT [MD = 3.94, 95% CI = 2.53 to 5.36, p-value < 0.00001] and [MD = 9.29, 95% CI = 0.67 to 17.91, p-value = 0.03] respectively. Moreover, the subgroup analysis showed that

**Funding:** The authors received no specific funding for this work.

**Competing interests:** The authors have declared that no competing interests exist.

the p-value of MDW levels [< 0.00001] is more significant than the p-value of PCT levels = 0.03, the p-value between the two subgroups [< 0.00001]. Additionally, the overall ROC Area for MDW [0.790] > the overall ROC Area for PCT [0.760].

## Conclusion

Our study revealed a statistically significant association between sepsis and increased MDW and PCT levels compared with controls and the overall ROC Area for MDW is higher than the overall ROC Area for PCT, indicating that the diagnostic accuracy of MDW is higher than PCT.MDW can be used as a diagnostic marker for sepsis patients in the emergency department. More multicenter studies are needed to support our findings.

## Introduction

Sepsis is a potentially fatal organ failure occurred by an inadequate host defense against infection [1, 2]. A systemic inflammatory reaction is how sepsis is typically described. In sepsis, the natural balance between pro- and anti-inflammatory activities becomes disrupted. Localized tissue reaction to infection becomes systemic, and the inflammatory process becomes itself becomes harmful [3]. It has three states: sepsis, severe sepsis, and septic shock, all of which advance with increasing severity [2]. Sepsis is one of the leading causes of mortality in intensive care units [ICUs] [4, 5]. Sepsis incidence rates are increasing, reaching 535 cases per 100,000 person-years [6]. Hospital mortality is still very high, at 25 to 30 percent [6].

Various biomarker subtypes have been identified based on their alleged applicability. Importantly, a single biomarker might fulfill several criteria for different applications, but it is crucial to establish a reliable justification for each explanation [7]. For instance, biomarkers based on monocytes are found to be involved in numerous inflammatory disorders. According to Tel et al, [8] type 2 diabetes mellitus patients in the frail and non-frail groups had significantly different monocyte lymphocyte ratio [MLR] levels during COVID-19. Frail individuals' values were noticeably greater than those of non-frail diabetics. MLR could therefore be used as a dependable and prognostic indicator of frailty [8]. Kocak et al. [9] measured the MRL in patients with diabetic kidney injury [DKI]. Two groups of T2DM patients—those with microalbuminuria [MA] and those with normoalbuminuria [NA]—were created. They stated that MLR and MA showed a statistically significant association. Thus, given its great association with MA, low cost, and availability, MLR might function as a predictive and efficient marker for DKI in diabetic people [9]. MRL was also investigated in the context of Covid-19 infection. According to reports, MLR increased in Covid-19 infection patients compared to healthy controls. To anticipate the disease, its severity, and the mortality of Covid-19 infection, MLR is a useful technique [10]. In patients with sepsis, different biomarkers have been investigated for application in infection identification, prognostication, and therapy guidance [11]. Sepsis biomarkers might offer information that isn't available through other parameters, this could help guide clinical decision-making and possibly enhance patient care. If the biomarkers that could precisely diagnose sepsis early were available, antibiotic therapy might be provided more promptly and appropriately and superfluous medications could be avoided. Similarly, biomarkers could assist healthcare providers in evaluating the effectiveness of their therapeutic decisions and modifying the treatment as necessary [12]. Numerous possible sepsis biomarkers have been presented. In particular, procalcitonin [PCT], C-reactive protein [CRP], white blood cells [WBC], erythrocyte sedimentation rate [ESR], and different interleukins have been

used as diagnostic biomarkers for SIRS, sepsis, and severe sepsis [13]. The clinical application of PCT and biomarkers is controversial. Yet, prior research found that PCT could support patient clinical diagnosis and care. Among these biomarkers, Surviving Sepsis Campaign 2013's diagnostic standards for inflammatory variables includes PCT and CRP [13]. Above all, early data on the potential importance of monocyte distribution width [MDW] were gathered [14], when activated in bacteremia patients, monocytes have been found to enlarge, and this infection-related size change can be easily observed by monitoring the distribution of monocytes in coulter chambers [14]. On the first visit, sepsis is frequently overlooked. Herein lies the value of MDW, whose results can be obtained sooner compared to those of other biomarkers. This offers a crucial study avenue to evaluate incorporating MDW within current routine WBC counts for sepsis detection [15]. Different results have been reported for the accuracy of PCT and MDW as diagnostic markers for sepsis, so we aimed to perform a meta-analysis to find out which one of them could be used as the accurate diagnostic marker for sepsis.

## Methods

This meta-analysis was performed according to Preferred Reporting Items for Systematic Reviews and Meta-Analyses [PRISMA] guidelines and the Cochrane handbook [16].

We searched PUBMED, Web of Science, and SCOPUS from inception to May 1, 2022. We used the following key terms;[["Monocyte Distribution Width"] OR ["Procalcitonin" or "Calcitonin Precursor Polyprotein" or "Calcitonin-1" or "Calcitonin Related Polypeptide Alpha"] AND ["Recognition of Sepsis"]]. We further reviewed the list of references of studies included in this meta-analysis to include other relevant studies. The studies were eligible for inclusion if they met the following criteria:

### Eligibility criteria

Any observational or clinical trials that compared monocyte Distribution Width [MDW] with Procalcitonin [PCT] as diagnostic markers in a patient with sepsis. Case reports, editorials, conference abstracts, and animal studies were excluded. The main outcomes were levels of MDW and PCT in the case group [sepsis] and control group [infection without sepsis] and sensitivity and specificity for MDW and PCT.

### Screening, data extraction, and risk of bias

Two reviewers [S.R and P.C] conducted initial title and abstract screening and all conflicts were discussed to reach an agreement, otherwise, a third opinion from [K.R.M] was obtained.

Potentially eligible articles were imported for full-text screening and assessed for inclusion. Data were extracted using an Excel sheet. Examples of data collected are study arms, number of patients in each group, age, sex [n], other baseline diseases, and baseline treatment.

We used the New Castle Ottawa Scale [NOS] tool to assess the quality of the included observational studies. Each study was ranked as; good, fair, or poor quality.

### Data analysis

Review Manager Software version 5.4 was used to perform the meta-analysis; the continuous outcomes were presented as mean difference [MD] with a 95% confidence interval. If heterogeneity was detected [Chi-square $P$ value $< 0.05$], a random effect model was used otherwise, we used a fixed-effect model, in general; the results were considered significant if the P-value was less than 0.05. MedCalc software was used to calculate the overall AUC for MDW and PCT.

## Results

### Literature search

After a comprehensive search of the literature, 768 studies resulted, and then became 663 were eligible for the title and abstract screening after the removal of duplicates. Of the 663, 654 were irrelevant and 9 studies were eligible for full-text screening. Finally, 5 observational studies were included in the meta-analysis after the full-text screening, as shown in the PRISMA in [Fig 1], summary of the included study is shown in Table 1.

MDW and PCT overall levels, MDW overall [AUC], and PCT overall [AUC] outcomes were compared between the sepsis group and control group. The overall quality was high in the included 5 studies as shown in Table 2.

The total number of patients included in the study is 1346 patients, 565 patients in the sepsis group, and 781 patients in the control group, other baseline data are shown in Table 3.

### Outcomes

**MDW and PCT overall levels.** The pooled analysis showed a statistically significant association between sepsis and increased levels of MDW compared with controls [MD = 3.94, 95%

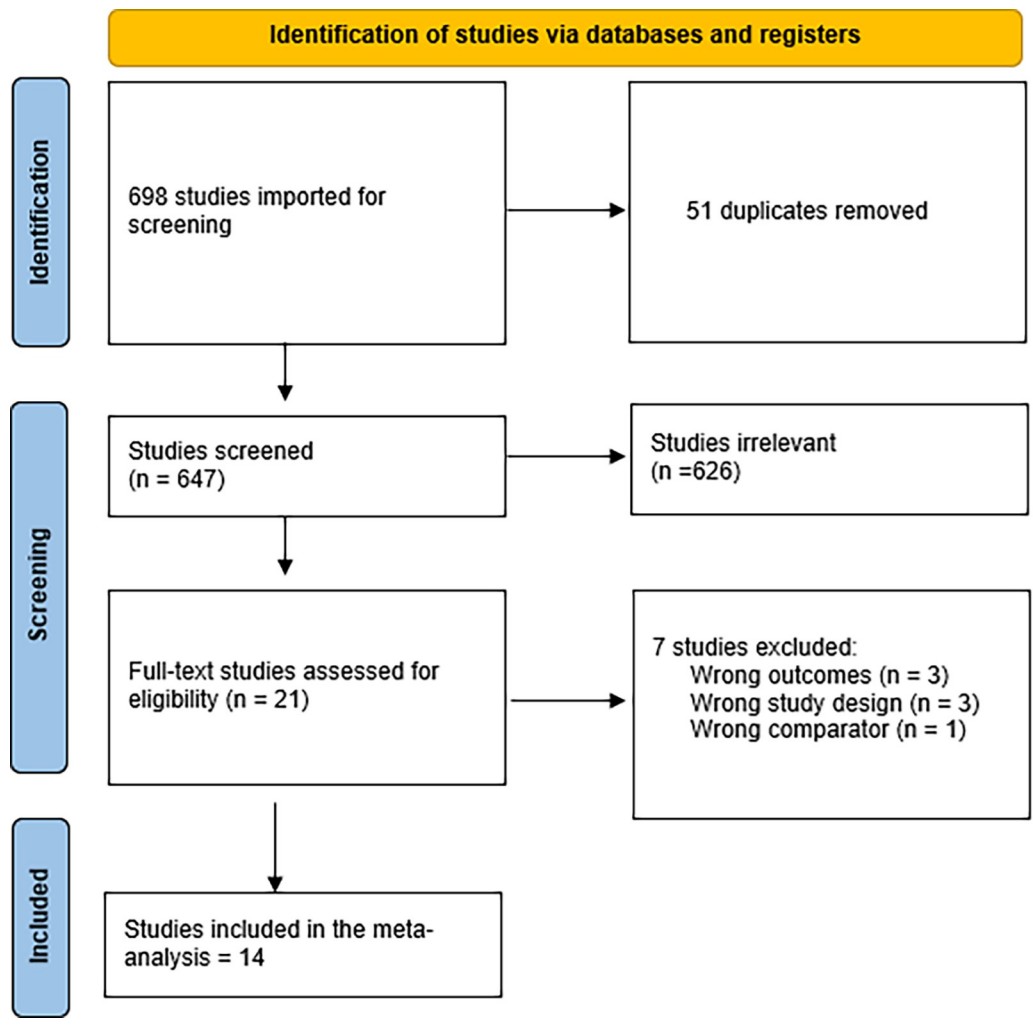

**Fig 1. PRISMA flow diagram.**

**Table 1. Summary of the included studies.**

| ID | Study design | Country of the study | Study arms | Outcomes | Conclusion |
|---|---|---|---|---|---|
| Polilli 2021 | Prospective cohort study | Italy | Case: a sample of consecutive patients assisted in an Intensive Care Unit for > 48 h with sepsis control: a sample of consecutive patients assisted in an Intensive Care Unit for > 48 h without sepsis | values of MDW > 23 were associated with a high PPV for sepsis, whereas values of MDW ≤ 20 were associated with a high NPV. The findings suggest that MDW may help clinicians to monitor ICU patients at risk of sepsis, with minimal additional efforts over standard of care | The study found that values of MDW > 23 were associated with a high PPV for sepsis, whereas values of MDW ≤ 20 were associated with a high NPV. Our findings suggest that MDW may help clinicians to monitor ICU patients at risk of sepsis, with minimal additional efforts over standard of care |
| Li 2022 | Prospective cohort study | Taiwan | Case: sepsis-3 groups Control: Infection without systemic infammatory response syndrome (SIRS), infection with SIRS | The AUC of MDW, PCT, and MDW +WBC to predict infection with SIRS was 0.753, 0.704, and 0.784, respectively (p<0.01). The sensitivity, specifcity, PPV, and NPV of MDW using 20 as the cutof were 86.4%, 54.2%, 76.4%, and 70%, compared to 32.9%, 88%, 82.5%, and 43.4% using 0.5 ng/mL as the PCT cutof value. On combing MDW and WBC count, the sensitivity and NPV further increased to 93.4% and 80.3%, respectively. In terms of predicting sepsis-3, the AUC of MDW, PCT, and MDW+WBC was 0.72, 0.73, and 0.70, respectively. MDW, using 20 as cutof, exhibited sensitiv-ity, specifcity, PPV, and NPV of 90.6%, 37.1%, 18.7%, and 96.1%, respectively, compared to 49.1%, 78.6%, 26.8%, and 90.6% when 0.5 ng/mL PCT was used as cutof. | In conclusion, MDW is a more sensitive biomarker than PCT in predicting infection-related SIRS and sepsis-3 in the ED. MDW<20 shows a higher NPV to exclude sepsis-3. Combining MDW and WBC count further improves the accuracy in predicting infection with SIRS but not sepsis-3. |
| Woo 2021 | Prospective cohort study | Korea | Case: Patients aged 18–80 years who visited the emergency department with sepsis Control: Patients aged 18–80 years who visited the emergency department with infection | The AUC values for MDW, CRP, PCT, and white blood cells for predicting sepsis were 0.71 (95% confidence interval [CI], 0.67–0.75), 0.75 (95% CI, 0.71–0.78], 0.76 (95% CI, 0.72–0.79, and 0.61 (95% CI, 0.57–0.65), respectively. With the optimal cutoff value of the cohort, the sensitivity was 83.0% for MDW (cutoff, 19.8), 69.7% for CRP (cutoff, 4.0), and 76.6% for PCT (cutoff, 0.05). The combination of quick Sequential Organ Failure Assessment (qSOFA) with MDW improved the AUC (0.76; 95% CI, 0.72–0.80) to a greater extent than qSOFA alone (0.67; 95% CI, 0.62–0.72). | MDW reflected a diagnostic performance comparable to that of conventional diagnostic markers, implying that MDW is an alternative biomarker. The combination of MDW and qSOFA improves the diagnostic performance for early sepsis. |

*(Continued)*

**Table 1.** (Continued)

| ID | Study design | Country of the study | Study arms | Outcomes | Conclusion |
|---|---|---|---|---|---|
| **Polilli 2020** | **Prospective cohort study** | italy | **Case: Patients with sepsis Control: Patients with bacteremia** | In multivariate models, MDW as a continuous variable (OR:1.57 for each unit increase; 95%CI: 1.31–1.87, p<0.001) and PCT>1 ng/ mL (OR: 48.5; 95%CI: 14.7–160.1, p<0.001) were independently associated with sepsis. Statistical best cut points associated with sepsis were 22.0 for MDW and 1.0 ng/mL for PCT whereas MDW values<20 were invariably associated with negative blood cultures. At ROC curve analysis, the AUC of MDW (0.87) was nearly overlapping that of PCT (0.88). | **The data suggest that incorporating MDW within current routine WBC counts and indices may be of remarkable use for detection of sepsis. Further research is warranted.** |
| **Hausfater 2021** | **Prospective cohort study** | **France and Spain** | **Case: Spesis 3 Control: Spesis 2** | MA total of 1,517 patients were analyzed: 837 men and 680 women, mean age 61 ±19 years, 260 (17.1%) categorized as Sepsis-2 and 144 patients (9.5%) as Sepsis-3. The AUCs [95% confdence interval] for the diagnosis of Sepsis-2 were 0.81 [0.78–0.84] and 0.86 [0.84–0.88] for MDW and MDW combined with WBC, respectively. For Sepsis-3, MDW performance was 0.82 [0.79–0.85]. The performance of MDW combined with WBC for Sepsis-2 in a subgroup of patients with low sepsis pretest probability was 0.90 [0.84–0.95]. The AUC for sepsis detection using MDW combined with WBC was similar to CRP alone (0.85 [0.83–0.87]) and exceeded that of PCT. Combining the biomarkers did not improve the AUC. Compared to normal MDW, abnormal MDW increased the odds of Sepsis-2 by factor of 5.5 [4.2–7.1, 95% CI] and Sepsis-3 by 7.6 [5.1–11.3, 95% CI]. | **MDW in combination with WBC has the diagnostic accuracy to detect sepsis, particularly when assessed in patients with lower pretest sepsis probability. We suggest the use of MDW as a systematic screening test, used together with qSOFA score to improve the accuracy of sepsis diagnosis in the emergency department** |

CI = 2.53 to 5.36, p-value < 0.00001]. We observed heterogeneity among studies [p-value < 0.00001, $I^2$ = 90%] Fig 2.

The pooled analysis showed a statistically significant association between sepsis and increased levels of PCT compared with controls [MD = 9.29, 95% CI = 0.67 to 17.91, p-value = 0.03]. We observed heterogeneity among studies [p-value < 0.00001, $I^2$ = 95%] Fig 2.

**Table 2. Quality assessment: The Newcastle Ottawa Scale [NOS] for assessing the quality of nonrandomized studies.**

| Study ID | The Newcastle Ottawa Scale [NOS] for assessing the quality of nonrandomized studies | | | Rating |
|---|---|---|---|---|
| | Selection | Comparability | Exposure [Outcome] | |
| **Ennio Polilli et al,.2021** | 4 Out of 4 | 2 Out of 2 | 3 Out of 3 | Good quality |
| **Chih-Huang Li et al,.2022** | 3 Out of 4 | 1 Out of 2 | 3 Out of 3 | Good quality |
| **A la Woo et al,.2021** | 4 Out of 4 | 1 Out of 2 | 3 Out of 3 | Good quality |
| **Ennio Polilli et al, 2020** | 4 Out of 4 | 2 Out of 2 | 3 Out of 3 | Good quality |
| **Pierre Hausfater et al, 2021** | 4 Out of 4 | 1 Out of 2 | 3 Out of 3 | Good quality |

**Table 3. Baseline characteristics of the included data.**

| ID | Number of patients in each group | | Age (Years) mean (SD) | | sex (n) | | | | Length of stay in the hospital or ICU days, mean (SD) | | Mortality (n) % | | Other baseline diseases (n) | |
| --- | --- | --- | --- | --- | --- | --- | --- | --- | --- | --- | --- | --- | --- | --- |
| | | | | | sepsis | | Infection | | | | | | | |
| | Sepsis | Infection | Sepsis | Infection | female | male | female | male | Sepsis | Infection | Sepsis | Infection | Sepsis | Infection |
| Polilli 2021 | 74 | 55 | 61.86 (16.06) | 63.11 (17.40) | 24 | 50 | 23 | 32 | 14.09 (9.96) | 11.49 (6.96) | 29 (39.2) | 11 (20.0) | Intracerebral haemorrhage (10) Cardiovascular failure (9) Polytrauma (10) Respiratory failure (8) Acute ischemic Stroke (4) Acute kidney failure (4) Head trauma (3) Brain surgery (2) Acidosis in metformin use (2) Septic shock (4) Haemorrhagic shock (3) Peritonitis (3) Acute pancreatitis (2) Coma in encephalitis (2) Consequence of Duodeno-cephalo-Pancreatectomy (1) Anaphylactic shock (1) | Intracerebral haemorrhage (14) Cardiovascular failure (10) Polytrauma (9) Respiratory failure (3) Acute ischemic Stroke (8) Acute kidney failure (1) Head trauma (3) Brain surgery (3) Acidosis in metformin use (1) |
| Li 2022 | 402 | | 63.7±18.9 | | Females = 201 Males = 201 | | | | | | 35 | | Diabetes (106) Hypertension (153) Chronic obstructive pulmonary disease (17) Chronic kidney disease (47) Congestive heart failure (13) Malignancy (119) Stroke (25) Liver cirrhosis (14) | |
| Woo 2021 | 188 | 132 | 63.4 (11.0) | 55.5 (13.9) | 78 | 110 | 58 | 74 | | | | | Malegnancy (117) HIV or organ transplant (3) chemotherapy (30) | Malignancy (71) chemotherapy (53) |
| Polilli 2020 | 105 | 155 | 65.4 (19.6) | 54.9 (18.3) | 41 | 64 | 54 | 101 | 13.4 (8.0) | 10.9 (8.4) | 17 (16.2) | 4 (2.6) | kidney disease (40) | kidney disease (20) |
| Hausfater 2021 | 144 | 260 | 72 (61–80) | 66 (51–77) | 46 | 98 | 101 | 159 | | | | | | |

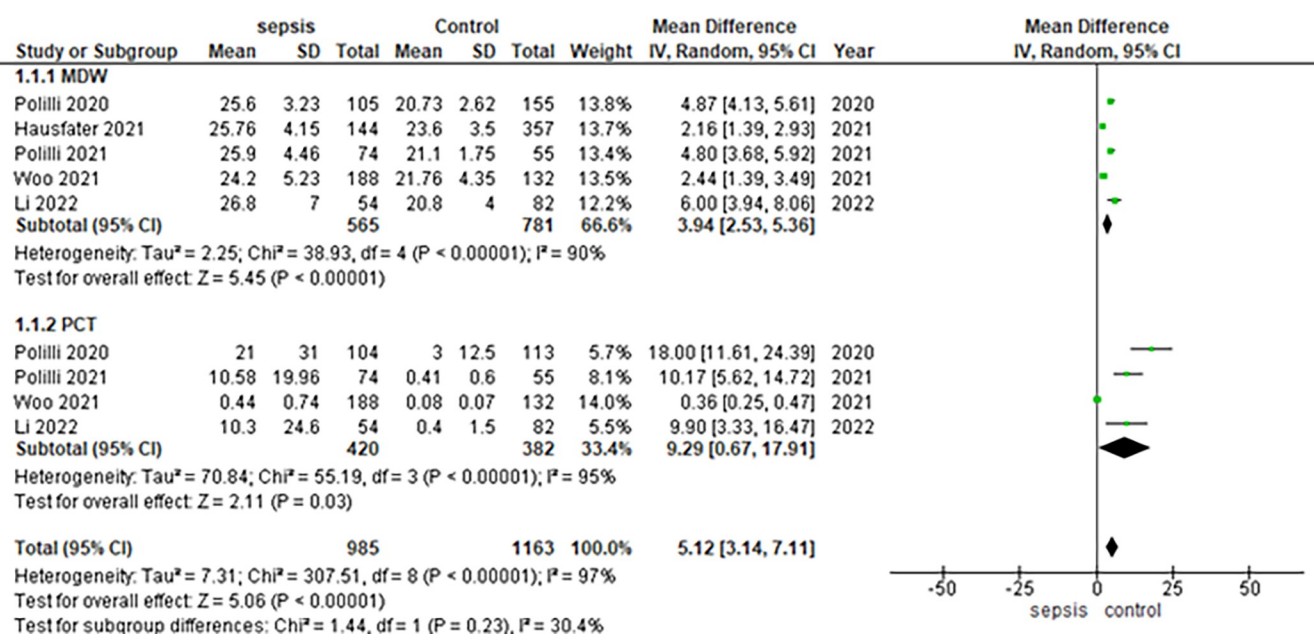

**Fig 2. Forest plot of the subgroup analysis between MDW and PCT levels.**

The subgroup analysis showed that the p-value of MDW levels [< 0.00001] is more significant than the p-value of PCT levels = 0.03, the p-value between the two subgroups [< 0.00001]. We observed no significant heterogeneity between the two subgroups [p-value = 0.23, $I^2$ = 30.4%], Fig 2.

**MDW overall [AUC].** The overall analysis showed that the pooled ROC Area for all 5 included studies was 0.790, 95% CI [0.732 to 0.848]. We detected significant heterogeneity among studies [p < 0.0001, $I^2$ = 87.30%], Fig 3.

**PCT overall [AUC].** The overall analysis showed that the pooled ROC Area for all 5 included studies was 0.760, 95% CI [0.681 to 0.840]. We detected significant heterogeneity among studies [p < 0.0001, $I^2$ = 90.52%], Fig 4. So the overall ROC Area for MDW [0.790] > the overall ROC Area for PCT [0.760], Fig 4.

**Sensitivity and specificity of MDW different levels [AUC].** The multiple test analysis showed that the pooled ROC Area for [MDW >20] > the ROC areas of the [MDW >22] and [MDW >20], Figs 5 and 6.

## Discussion

Our meta-analysis showed a statistically significant association between sepsis and increased MDW and PCT levels compared with controls, the subgroup analysis showed that the p-value of MDW is more significant than the p-value of PCT in sepsis diagnosis, another analysis showed that the overall ROC Area for MDW is higher than the overall ROC Area for PCT, indicating that the diagnostic accuracy of MDW is higher than PCT.

The multiple test analysis showed that the pooled ROC Area for [MDW >20] > the ROC areas of the [MDW >22] and [MDW >20].

Blood culture is a crucial diagnostic tool for sepsis because it may be used to identify pathogenic bacteria and test for antibiotic sensitivity, however it's time-consuming and prone to false-negative results, especially after antibiotic usage [17]. According to statistics, people with sepsis have a survival rate of more than 80% if they are accurately diagnosed and treated within

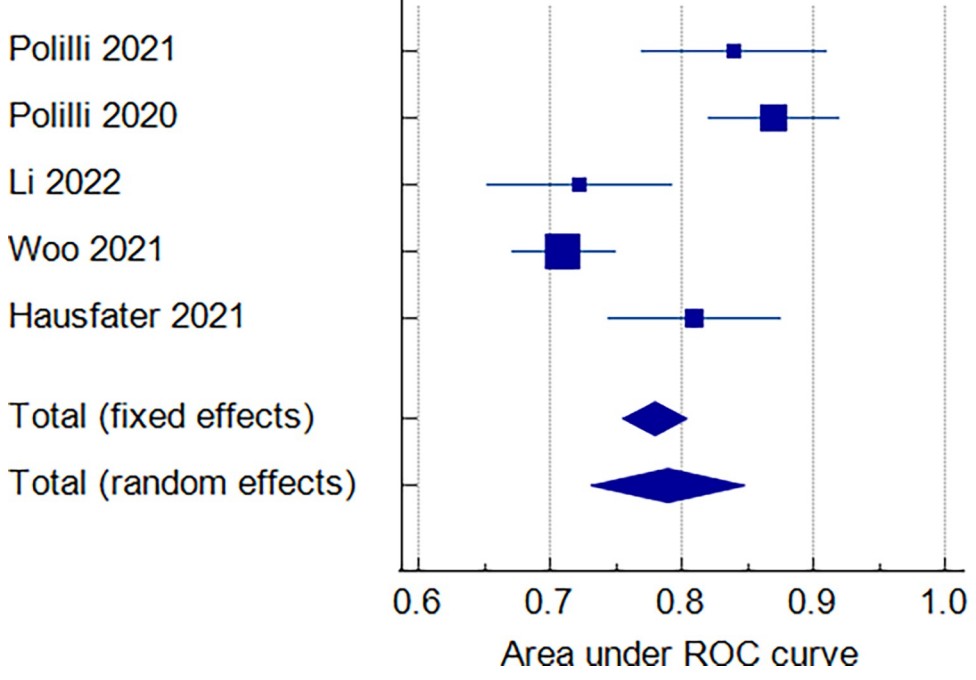

**Fig 3. Forest plot of the MDW [AUC].**

1 hour of infection, however this number reduces to 30% if they receive treatment after 6 hours [18]. Finding a biomarker for the early detection of sepsis is therefore essential. The molecules indicating changes in immunoinflammatory state, such as PCT and MDW, appear to be promising among the several sepsis biomarkers explored so far [19, 20].

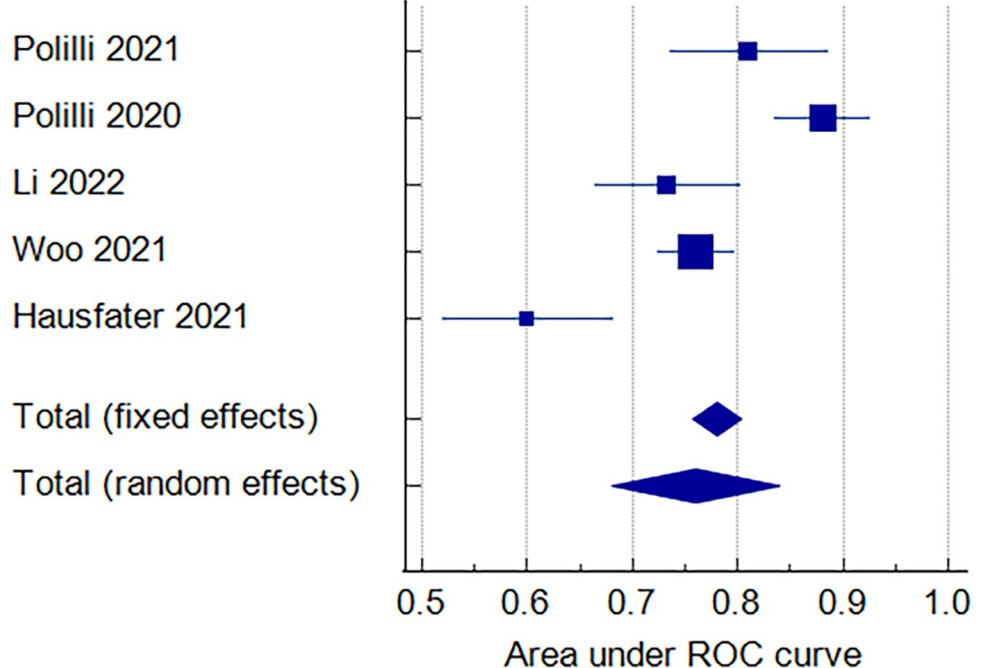

**Fig 4. Forest plot of the PCT [AUC].**

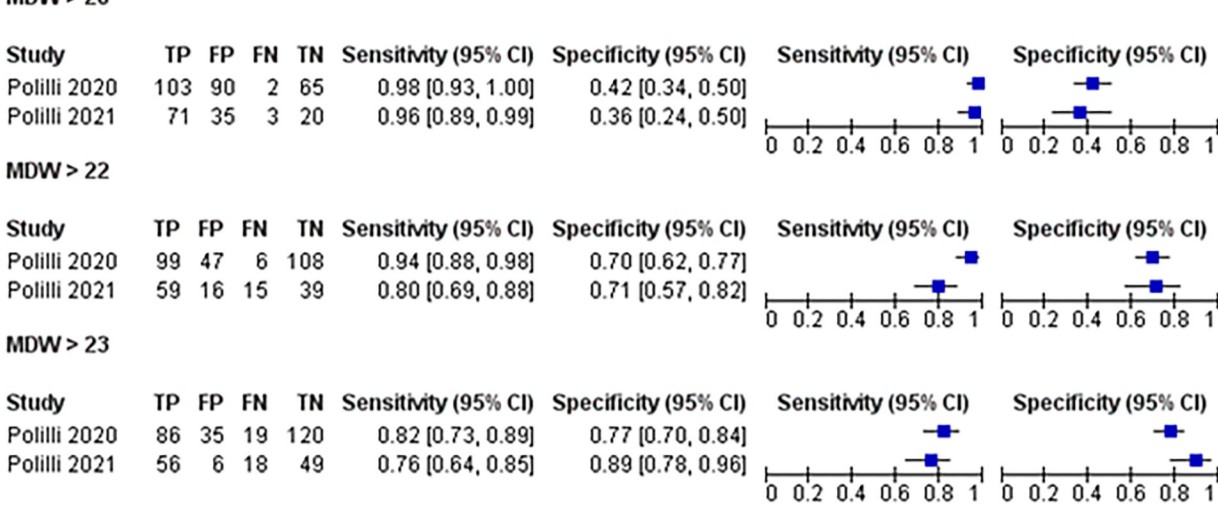

**Fig 5. Forest plot of the sensitivity and specificity of MDW different levels.**

PCT, which is produced by C cells in the thyroid gland and, to a lesser extent, other neuro-endocrine cells all throughout the body, is a precursor to the hormone calcitonin [21]. In very early stages of fetal development, it is also expressed in the central nervous system [22]. With a half-life of roughly 20–24 h, PCT is an extremely stable protein both in vitro and in vivo [23, 24]. Serum PCT levels in healthy people are typically very low [0.02 ng/mL] [18]. There are, however, two primary pathways to PCT production in the context of bacterial infection: the direct pathway, induced by LPS or other harmful microbial metabolites [25], and the indirect pathway, induced by proinflammatory cytokines [TNF-, IL-1, and IL-6] that cause the expression of CALC-1, the gene responsible for PCT production, in numerous cells all over the body [22]. Because PCT cannot be converted to calcitonin, it reaches the circulation system where its levels can quickly rise by more than 400 times [> 4.0 ng/mL] over baseline values [26]. Moreover, PCT should be consistently produced in both immunocompetent and immuno-compromised patients because it is produced by tissues in the setting of bacterial infection in addition to immune cells [22]. Furthermore, in that study, PCT performed better as a diagnostic biomarker for sepsis than C-reactive protein [CRP], IL-6, and lactate [27].

MDW, a hematologic parameter which is assessed as part of the complete blood count with differential [CBC-DIFF] [28], and measures the dispersion around the mean of the monocyte volume population in whole blood, could be an early indicator of infection and sepsis, according to some studies [14, 29–33]. MDW can be used to measure monocyte activity and morphological alterations during the early inflammatory response [32]. MDW role in sepsis detection in the ED first investigated by Crouser et al., demonstrating that MDW enhanced initial sepsis detection [32]. In our meta-analysis, we contrasted MDW with PCT, the most prevalent inflammatory biomarker used to detect sepsis.

Li et al compared MDW against PCT as diagnostic markers for sepsis patients, when they used 20 as the MDW cut-off value, significantly higher sensitivity and NPV were noticed with MDW than PCT at 0.5 ng/mL as the cut-off value. Also, at the MDW cut-off value of 20, an excellent negative prognostic value for sepsis exclusion was revealed. The Overall results showed that MDW had better predictive value in both early sepsis screening and patient outcome [34].

Woo et al compared MDW against PCT and CRP as diagnostic markers for sepsis patients, MDW revealed a higher sensitivity than other markers at optimal cutoff from the cohort

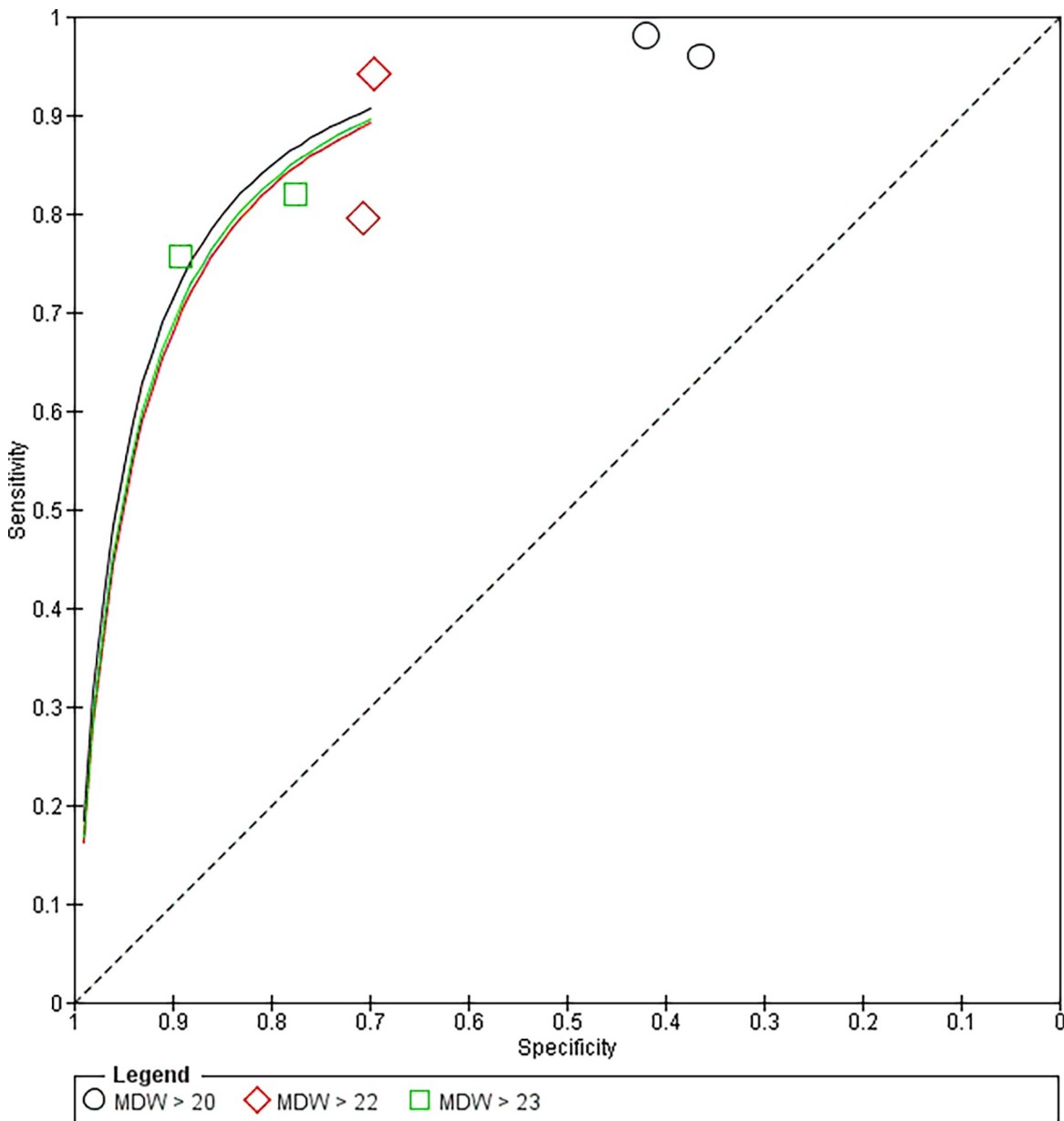

**Fig 6. SROC plot of the sensitivity and specificity of MDW different levels.**

which is explained by the rapid response of monocyte to infection. The analysis also showed that MDW revealed a significantly higher diagnostic accuracy in terms of AUC in the overall population by adding clinical score [qSOFA], in contrast to adding tests for CRP and PCT which do not show advantages in diagnosis of sepsis [35].

Polilli et al compared MDW against PCT as diagnostic markers for sepsis patients, they found that the MDW cutoff of 21.9 was shown to be the most accurate threshold for sepsis prediction, so for sepsis prediction, MDW at >22 was extremely sensitive and specific. They also found that Low MDW values may be a useful tool to rule out bloodstream infections because they had a very high NPV for sepsis and a 100% NPV for bacteremia at MDW values <20 which could be used to rule out sepsis in routine blood counts [14]. Polilli et al 2021 compared

MDW against PCT as diagnostic markers for sepsis patients, they found that using 20 as the MDW cut-off value was associated highest levels of NPV [86.4%] and sensitivity [95.9%], while combined MDW > 23 and PCT > 0.5 ng/mL were associated with the highest specificity [92.6%], Youden Index [0.61] and PPV [92.6%. which suggests using both MDW and PCT as optimal rules in sepsis prediction for ICU patients [36]. Our results agree with the results of Pollilio 2020 and Polillio 2021 which reported that MDW>20 has more sensitivity and specificity and has more diagnostic value than other levels [14, 36].

One explanation for the enhanced sensitivity of MDW in the sepsis diagnosis may be the fact that monocyte differentiation in the circulation starts relatively early in the sepsis cascade. One of the limitations of MDW is that in patients with a monocyte event <100 count in the peripheral blood sample, the MDW value is not available [34]. Clinically it is difficult to exclude critical patients using PCT as PCT was found to be a specific but not a sensitive biomarker in terms of sepsis-3 prediction [37, 38]. The results of our study can be used in clinical practice by depending on MDW more than PCT as a diagnostic marker in sepsis patients and MDW>20 should be considered the cut-off value for the diagnosis of sepsis in intensive care units. Our study is strengthened by the high sample size [1346 patients], the overall quality was good in all 5 included studies, also it is the first met analysis of diagnostic test accuracy studies to compare the diagnostic accuracy of MDW against PCT for sepsis, we also compared the accuracy of the two diagnostic markers in the same patients. However, our study is limited by the detected heterogeneity among studies but the overall heterogeneity between MDW and PCT subgroups was not significant.

## Future implications

Our analysis results revealed that PCT and MDW can be used as diagnostic markers for sepsis detection, with higher diagnostic accuracy of MDW than PCT. Additionally, MDW cut-off value of 20, can be used as an excellent negative prognostic value for sepsis exclusion.

## Conclusion

Our study revealed a statistically significant association between sepsis and increased MDW and PCT levels compared with controls and the overall ROC Area for MDW is higher than the overall ROC Area for PCT, indicating that the diagnostic accuracy of MDW is higher than PCT. MDW markers with a cut-off value > 20 could be used as diagnostic markers of sepsis patients in the emergency department. More multicenter studies are needed to support our findings.

## Supporting information

**S1 Checklist. PRISMA 2020 checklist.**
(DOCX)

## Author Contributions

**Conceptualization:** Samah S. Rozan.

**Data curation:** Samah S. Rozan.

**Supervision:** Karam R. Motawea, Nesreen Elsayed Talat, Hani Aiash.

**Validation:** Karam R. Motawea.

**Writing – original draft:** Nesreen Elsayed Talat, Rowan H. Elhalag, Sarraa Mohammed Reyad, pensée chebl, Sarya Swed, Bisher Sawaf, Hadeel Hadeel alfar, Amr Farwati, Bana Sabbagh, Esperance M. Madera, Amro El Metaafy.

**Writing – review & editing:** Rowan H. Elhalag, Sarraa Mohammed Reyad, pensée chebl, Sarya Swed, Bisher Sawaf, Hadeel Hadeel alfar, Amr Farwati, Bana Sabbagh, Esperance M. Madera, Amro El Metaafy, Joshuan J. Barboza, Ranjit Sah.

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
