## [Decision Letter · Decision Letter 0]

6 Mar 2023

PONE-D-22-35604Comparison of Monocyte distribution width and Procalcitonin as diagnostic markers for sepsis: Meta-analysis of diagnostic test accuracy studiesPLOS ONE

Dear Dr. Swed,

Thank you for submitting your manuscript to PLOS ONE. After careful consideration, we feel that it has merit but does not fully meet PLOS ONE’s publication criteria as it currently stands. Therefore, we invite you to submit a revised version of the manuscript that addresses the points raised during the review process.

One of the reviewers suggest rejection and the other recommend major revision. Issues mentioned by the reviewer who suggest rejection ca be fixed by an extensive revision. Therefore, I think the paper can be subjected to a decision of revision. Especially be careful in revising the issues mentioned by reviewer 1. The paper can not be further processed unless adequately revised.

We look forward to receiving your revised manuscript.

Kind regards,

Gulali Aktas

Academic Editor

PLOS ONE

Journal Requirements:

https://pubmed.ncbi.nlm.nih.gov/31420921/

In your revision ensure you cite all your sources (including your own works), and quote or rephrase any duplicated text outside the methods section. Further consideration is dependent on these concerns being addressed.

    "N/a"

     "N/a"

6. Please include a copy of Table 1, 2 and 3 which you refer to in your text on page 11.

Additional Editor Comments:

One of the reviewers suggest rejection and the other recommend major revision. Issues mentioned by the reviewer who suggest rejection ca be fixed by an extensive revision. Therefore, I think the paper can be subjected to a decision of revision.

Reviewers' comments:

Reviewer's Responses to Questions

**Comments to the Author**

1. Is the manuscript technically sound, and do the data support the conclusions?

Reviewer #1: No

Reviewer #2: Yes

2. Has the statistical analysis been performed appropriately and rigorously? 

Reviewer #1: No

Reviewer #2: Yes

3. Have the authors made all data underlying the findings in their manuscript fully available?

Reviewer #1: No

Reviewer #2: Yes

4. Is the manuscript presented in an intelligible fashion and written in standard English?

Reviewer #1: No

Reviewer #2: Yes

5. Review Comments to the Author

Reviewer #1: There are too many typos in the text.Some data in sections and tables are not compatible with each other. There is no table, there are only figures.Figure numbers should be checked. Statistical analyses are incomplete.Therefore,results are unreliable.Discussion is far from being adequate. References are too few such as this topic and these major issues cannot be revised even with extensive check. Thus I recommend against publication.

Reviewer #2: Logic of the study must be expressed better in introduction. Why authors studied mdw in sepsis. Sepsis is associated with high inflammatory burden. Monicyte based markers are also associated with inflammatory diseases including diabetic nephropathy (Journal of Diabetes & Metabolic Disorders, 19, 997-1002.), frailty (Clinical Diabetology, 2022, 11.2: 119-126.), and other infections (Revista da Associação Médica Brasileira, 2021, 67: 1-2.).

İn discussion,discuss possible clinical usage of the results of the present study.

6. PLOS authors have the option to publish the peer review history of their article (what does this mean?). If published, this will include your full peer review and any attached files.

Reviewer #1: No

Reviewer #2: No

---

## [Author Response · Author response to Decision Letter 0]

24 May 2023

Response to the handling editor

Response: we have edited the style of the manuscript to be same the formal style in PLOSE ONE journal 

https://pubmed.ncbi.nlm.nih.gov/31420921/

In your revision ensure you cite all your sources (including your own works), and quote or rephrase any duplicated text outside the methods section. Further consideration is dependent on these concerns being addressed.

Response: we have paraphrased the similar phrases 

"N/a"

Response: we confirm that there is no external fund for our paper 

"N/a"

Response: we confirm that there is no conflict of interest 

Response: Data Availability; as the study is review article; the analyzed data is included within the manuscript.

6. Please include a copy of Table 1, 2 and 3 which you refer to in your text on page 11.

Response: Done 

Response: Done 

8. Can you please upload an additional copy of your revised manuscript that does not contain any tracked changes or highlighting as your main article file. This will be used in the production process if your manuscript is accepted. Please amend the file type for the file showing your changes to Revised Manuscript w/tracked changes. Please follow this link for more information: http://blogs.PLOS.org/everyone/2011/05/10/how-to-submit-your-revised-manuscript/

Response: Done

Response to reviewers:

Reviewer #1: 

-There are too many typos in the text.

*We have edited the language and corrected the typos.

-Some data in sections and tables are not compatible with each other. There is no table, there are only figures. 

*We haved added the tables as separate files to the submission files and made it compatible with data in the manuscript. 

-Figure numbers should be checked. 

*We have edited the number of figures in results and figures section. 

-Statistical analyses are incomplete.Therefore,results are unreliable.

*We did the analysis according to data reported in the included studies. We could not include other outcomes and cutoff values because they were reported once in the studies, so we could not pool them in the results. We were restricted with the outcomes and analysis in our study because they were reported in more than one study , so we could pool them in the analysis.

-Discussion is far from being adequate. References are too few such as this topic and these major issues cannot be revised even with extensive check. Thus I recommend against publication.

*We have updated the discussion with more references, about 20 references related to our topic as shown in the discussion. 

Reviewer #2: 

-Logic of the study must be expressed better in introduction. Why authors studied mdw in sepsis. Sepsis is associated with high inflammatory burden. Monicyte based markers are also associated with inflammatory diseases including diabetic nephropathy (Journal of Diabetes & Metabolic Disorders, 19, 997-1002.), frailty (Clinical Diabetology, 2022, 11.2: 119-126.), and other infections (Revista da Associação Médica Brasileira, 2021, 67: 1-2.).

*We have updated the introduction with your modifications and added your suggested references. The edits are marked with yellow in the introduction.

-İn discussion,discuss possible clinical usage of the study.

*We have edited the discussion and added the possible clinical usage of the study at the end of the discussion before conclusion.

---

## [Decision Letter · Decision Letter 1]

21 Jun 2023

Comparison of Monocyte distribution width and Procalcitonin as diagnostic markers for sepsis: Meta-analysis of diagnostic test accuracy studies

PONE-D-22-35604R1

Dear Dr. Swed,

We’re pleased to inform you that your manuscript has been judged scientifically suitable for publication and will be formally accepted for publication once it meets all outstanding technical requirements.

Kind regards,

Gulali Aktas

Academic Editor

PLOS ONE

Additional Editor Comments (optional):

Reviewers' comments:

Reviewer's Responses to Questions

**Comments to the Author**

1. If the authors have adequately addressed your comments raised in a previous round of review and you feel that this manuscript is now acceptable for publication, you may indicate that here to bypass the “Comments to the Author” section, enter your conflict of interest statement in the “Confidential to Editor” section, and submit your "Accept" recommendation.

Reviewer #1: All comments have been addressed

Reviewer #2: All comments have been addressed

2. Is the manuscript technically sound, and do the data support the conclusions?

Reviewer #1: Yes

Reviewer #2: Yes

3. Has the statistical analysis been performed appropriately and rigorously? 

Reviewer #1: Yes

Reviewer #2: Yes

4. Have the authors made all data underlying the findings in their manuscript fully available?

Reviewer #1: Yes

Reviewer #2: Yes

5. Is the manuscript presented in an intelligible fashion and written in standard English?

Reviewer #1: Yes

Reviewer #2: Yes

6. Review Comments to the Author

Reviewer #1: (No Response)

Reviewer #2: Dear Editör

the authors provided the corrections appropriately article should be accepted.

7. PLOS authors have the option to publish the peer review history of their article (what does this mean?). If published, this will include your full peer review and any attached files.

Reviewer #1: No

Reviewer #2: No

---

## [Editor Report · Acceptance letter]

26 Jul 2023

PONE-D-22-35604R1 

Comparison of Monocyte distribution width and Procalcitonin as diagnostic markers for sepsis: Meta-analysis of diagnostic test accuracy studies 

Dear Dr. Swed:

I'm pleased to inform you that your manuscript has been deemed suitable for publication in PLOS ONE. Congratulations! Your manuscript is now with our production department. 

Kind regards, 

on behalf of

Professor Gulali Aktas 

Academic Editor

PLOS ONE